# Sharp Bounds for Generalized Uniformity Testing

**Ilias Diakonikolas**
University of Southern California
diakonik@usc.edu

**Daniel M. Kane**
University of California, San Diego
dakane@ucsd.edu

**Alistair Stewart**
University of Southern California
stewart.al@gmail.com

## Abstract

We study the problem of *generalized uniformity testing* of a discrete probability distribution: Given samples from a probability distribution $p$ over an *unknown size* discrete domain $\Omega$, we want to distinguish, with probability at least $2/3$, between the case that $p$ is uniform on some *subset* of $\Omega$ versus $\epsilon$-far, in total variation distance, from any such uniform distribution. We establish tight bounds on the sample complexity of generalized uniformity testing. In more detail, we present a computationally efficient tester whose sample complexity is optimal, within constant factors, and a matching worst-case information-theoretic lower bound. Specifically, we show that the sample complexity of generalized uniformity testing is $\Theta\left(1/(\epsilon^{4/3}\|p\|_3) + 1/(\epsilon^2\|p\|_2)\right)$.

## 1 Introduction

Consider the following statistical task: Given independent samples from a distribution over an *unknown size* discrete domain $\Omega$, determine whether it is uniform on some *subset* of the domain versus significantly different from any such uniform distribution. Formally, let $\mathcal{C}_U \overset{\text{def}}{=} \{\mathbf{u}_S : S \subseteq \Omega\}$ denote the set of uniform distributions $\mathbf{u}_S$ over subsets $S$ of $\Omega$. Given sample access to an unknown distribution $p$ on $\Omega$ and a proximity parameter $\epsilon > 0$, we want to correctly distinguish between the case that $p \in \mathcal{C}_U$ versus $d_{\mathrm{TV}}(p, \mathcal{C}_U) \overset{\text{def}}{=} \min_{S \subseteq \Omega} d_{\mathrm{TV}}(p, \mathbf{u}_S) \geq \epsilon$, with probability at least $2/3$. Here, $d_{\mathrm{TV}}(p, q) = (1/2)\|p - q\|_1$ denotes the total variation distance between distributions $p$ and $q$. This natural problem, termed *generalized uniformity testing*, was recently introduced by Batu and Canonne [BC17], who gave the first upper and lower bounds on its sample complexity.

Generalized uniformity testing bears a strong resemblance to the familiar task of *uniformity testing*, where one is given samples from a distribution $p$ on a domain of *known size* $n$ and the goal is to determine, with probability at least $2/3$, whether $p$ is the uniform distribution $\mathbf{u}_n$ on this domain versus $d_{\mathrm{TV}}(p, \mathbf{u}_n) \geq \epsilon$. Uniformity testing is arguably *the* most extensively studied problem in distribution property testing [GR00, Pan08, VV14, DKN15b, Gol16, DGPP16, DGPP17] and its sample complexity is well understood. Specifically, it is known [Pan08, CDVV14, VV14, DKN15b] that $\Theta(n^{1/2}/\epsilon^2)$ samples are necessary and sufficient for this task.

The field of *distribution property testing* [BFR+00] has seen substantial progress in the past decade, see [Rub12, Can15] for two recent surveys. A large body of the literature has focused on characterizing the sample size needed to test properties of arbitrary distributions of a *given* support size. This regime is fairly well understood: for many properties of interest there exist sample-efficient testers [Pan08, CDVV14, VV14, DKN15b, ADK15, CDGR16, DK16, DGPP16, CDS17, DGPP18, CDKS18]. Moreover, an emerging body of work has focused on leveraging a pri-

ori structure of the underlying distributions to obtain significantly improved samples complexities [BKR04, DDS+13, DKN15b, DKN15a, CDKS17, DP17, DDK16, DKN17].

Perhaps surprisingly, the natural setting where the distribution is arbitrary on a discrete but unknown domain (of unknown size) does not seem to have been explicitly studied before the recent work of Batu and Canonne [BC17]. Returning to the specific problem studied here, at first glance it might seem that generalized uniformity testing and uniformity testing are essentially the same task. Naively, one might attempt to apply the existing uniformity testers directly without explicit knowledge of the domain. This nearly works, as standard testers do not need to make use of any particular information about the names of domain elements. However, these algorithms do make use of the *domain size* in a critical way. This difficulty is not so easy to overcome. In fact, as was shown in [BC17], the sample complexity with an unknown domain size is significantly different. Specifically, [BC17] gave a generalized uniformity tester with expected sample complexity $O(1/(\epsilon^6 \|p\|_3))$ and showed a lower bound of $\Omega(1/\|p\|_3)$. This should be compared to the $O(n^{1/2}/\epsilon^2)$-sample tester for distributions on domains of size $n$. Of particular interest here is that distributions $p$ with support size $n$ can have $1/\|p\|_3$ as large as $n^{2/3}$, making the problem with unknown domain substantially harder in the worst case.

## 1.1 Our Results and Techniques

An immediate open question arising from the work of [BC17] is to precisely characterize the sample complexity of generalized uniformity testing. The main result of this paper provides an answer to this question. In particular, we show the following:

**Theorem 1.1** (Main Result). *There is an algorithm with the following performance guarantee: Given sample access to an arbitrary distribution $p$ over an unknown size discrete domain $\boldsymbol{\Omega}$ and a parameter $0 < \epsilon < 1$, the algorithm uses $O\left(1/(\epsilon^{4/3}\|p\|_3) + 1/(\epsilon^2\|p\|_2)\right)$ independent samples from $p$ in expectation, and distinguishes between the case $p \in \mathcal{C}_U$ versus $d_{\mathrm{TV}}(p, \mathcal{C}_U) \geq \epsilon$ with probability at least $2/3$. Moreover, for every $0 < \epsilon < 1/10$ and $n > 1$, any algorithm that distinguishes between $p \in \mathcal{C}_U$ and $d_{\mathrm{TV}}(p, \mathcal{C}_U) \geq \epsilon$ requires at least $\Omega(n^{2/3}/\epsilon^{4/3} + n^{1/2}/\epsilon^2)$ samples, where $p$ is guaranteed to have $\|p\|_3 = \Theta(n^{-2/3})$ and $\|p\|_2 = \Theta(n^{-1/2})$.*

In the following paragraphs, we provide an intuitive explanation of our algorithm and our matching sample size lower bound, in tandem with a comparison to the prior work [BC17].

**Sample-Optimal Generalized Uniformity Tester**   Our algorithm requires considering two cases based on the relative size of $\epsilon$ and $\|p\|_2^2$. This case analysis seems somewhat intrinsic to the problem as the correct sample complexity branches into these cases.

For large $\epsilon$, we use the same overall technique as [BC17], noting that $p$ is uniform if and only if $\|p\|_3 = \|p\|_2^{4/3}$, and that for $p$ far from uniform, $\|p\|_3$ must be substantially larger. The basic idea from here is to first obtain rough approximations to $\|p\|_2$ and $\|p\|_3$ in order to ascertain the correct number of samples to use, and then use standard unbiased estimators of $\|p\|_2^2$ and $\|p\|_3^3$ to approximate them to appropriate precision, so that their relative sizes can be compared with appropriate accuracy.

We improve upon the work of [BC17] in this parameter regime in a couple of ways. First, we obtain more precise lower bounds on the difference $\|p\|_3^3 - \|p\|_2^4$ in the case where $p$ is far from uniform (Lemma 2.8). This allows us to reduce the accuracy needed in estimating $\|p\|_2$ and $\|p\|_3$. Second, we refine the method used for performing the approximations to these moments ($\ell_r$-norms). In particular, we observe that using the generic estimators for these quantities yields sub-optimal bounds for the following reason: The error of the unbiased estimators is related to their variance, which in turn can be expressed in terms of the higher moments of $p$ (Fact 2.1). This implies for example that the worst case sample complexity for estimating $\|p\|_3$ comes when the fourth and fifth moments of $p$ are large. However, since we are trying to test for the case of uniformity (where these higher moments are minimal), we do not need to worry about this worst case. In particular, after applying sample efficient tests to ensure that the higher moments of $p$ are not much larger than expected, the standard estimators for the second and third moments of $p$ can be shown to converge more rapidly than they would in the worst case (Fact 2.1).

The above algorithm is not sufficient for small values of $\epsilon$. For $\epsilon$ sufficiently small, we employ a different, perhaps more natural, algorithm. Here we take $m$ samples (for $m$ appropriately chosen based on an approximation to $\|p\|_2$) and consider the subset $S$ of the domain that appears in the sample. We then test whether the conditional distribution $p$ on $S$ is uniform, and output the answer of this tester. The number of samples $m$ drawn in the first step is sufficiently large so that $p(S)$, the probability mass of $S$ under $p$, is relatively high. Hence, it is easy to sample from the conditional distribution using rejection sampling. Furthermore, we can use a standard uniformity testing algorithm requiring $O(\sqrt{|S|}/\epsilon^2)$ samples.

To establish correctness of this algorithm, we need to show that if $p$ is far from uniform, then the conditional distribution $p$ on $S$ is far from uniform as well. We show (Lemma 2.10) that for any $x = \Theta(1/n)$, with high constant probability, the random variable $Z(x) = \sum_{i \in S} |p_i - x|$ is large. It is not hard to show that this holds with high probability for each fixed $x$, as $p$ being far from uniform implies that $\sum_{i \in \mathbf{\Omega}} \min(p_i, |p_i - x|)$ is large. This latter condition can be shown to provide a clean lower bound for the expectation of $Z(x)$. To conclude the argument, we show that $Z(x)$ is tightly concentrated around its expectation. Applying an appropriate union bound, allows us to show that $Z(x)$ is large for *all* $x$, and thus that the conditional distribution is far form uniform.

**Sample Complexity Lower Bound** The lower bound of $\Omega(n^{1/2}/\epsilon^2)$ follows directly from the standard lower bound of [Pan08] for uniformity testing on a given domain of size $n$. The other branch of the lower bound, namely $\Omega(n^{2/3}/\epsilon^{4/3})$, is more involved. To prove this lower bound, we use the shared information method of [DK16] for the following family of hard instances: In the "YES" case, we consider the distribution over (pseudo-)distributions on $N$ bins, where each $p_i$ is $(1+\epsilon^2)/n$ with probability $n/(N(1+\epsilon^2))$, and 0 otherwise. (Here we assume that the parameter $N$ is sufficiently large compared to the other parameters.) In the "NO" case, we consider the distribution over (pseudo-)distributions on $N$ bins, where each $p_i$ is $(1+\epsilon)/n$ with probability $n/(2N)$, $(1-\epsilon)/n$ with probability $n/(2N)$, and 0 otherwise.

**Notation.** Let $\mathbf{\Omega}$ denote the unknown discrete domain. Each probability distribution over $\mathbf{\Omega}$ can be associated with a probability mass function $p : \mathbf{\Omega} \to \mathbb{R}_+$ such that $\sum_{i \in \mathbf{\Omega}} p_i = 1$. We will use $p_i$, instead of $p(i)$, to denote the probability of element $i \in \mathbf{\Omega}$ in $p$. For a distribution $p$ and a set $S \subseteq \mathbf{\Omega}$, we denote by $p(S) \stackrel{\text{def}}{=} \sum_{i \in S} p_i$ and by $(p|S)$ the conditional distribution of $p$ on $S$. For $r \geq 1$, the $\ell_r$-norm of a function $p : \mathbf{\Omega} \to \mathbb{R}$ is $\|p\|_r \stackrel{\text{def}}{=} \left( \sum_{i \in \mathbf{\Omega}} |p_i|^r \right)^{1/r}$. For convenience, we will denote $\mathbf{F}_r(p) \stackrel{\text{def}}{=} \|p\|_r^r = \sum_{i \in \mathbf{\Omega}} |p_i|^r$. For $\emptyset \neq S \subseteq \mathbf{\Omega}$, let $\mathbf{u}_S$ be the uniform distribution over $S$. Let $\mathcal{C}_U \stackrel{\text{def}}{=} \{\mathbf{u}_S : \emptyset \neq S \subseteq \mathbf{\Omega}\}$ be the set of uniform distributions over subsets of $\mathbf{\Omega}$. The total variation distance between distributions $p, q$ on $\mathbf{\Omega}$ is defined as $d_{\mathrm{TV}}(p, q) \stackrel{\text{def}}{=} \max_{S \subseteq \mathbf{\Omega}} |p(S) - q(S)| = (1/2) \cdot \|p - q\|_1$. Finally, we denote by $\mathrm{Poi}(\lambda)$ the Poisson distribution with parameter $\lambda$.

## 2 Generalized Uniformity Tester

Before we describe our algorithm, we summarize a few preliminary results on estimating the power sums $\mathbf{F}_r(p) = \sum_{i \in \mathbf{\Omega}} |p_i|^r$ of an unknown distribution $p$. We present these results in Section 2.1. In Section 2.2, we present and analyze the algorithm for large values of $\epsilon$. In Section 2.3, we do the same for the small $\epsilon$ algorithm. Finally, in Section 2.4, we present the full algorithm.

### 2.1 Estimating the Power Sums of a Discrete Distribution

We will require various notions of approximation for the power sums of a discrete distribution.

**Fact 2.1** ([AOST17])**.** *Let $p$ be a probability distribution on an unknown discrete domain. For any $r \geq 1$, there exists an estimator $\widehat{\mathbf{F}}_r(p)$ for $\mathbf{F}_r(p)$ that draws $\mathrm{Poi}(m)$ samples from $p$ and satisfies the following: $\mathbf{E}[\widehat{\mathbf{F}}_r(p)] = \mathbf{F}_r(p)$ and $\mathbf{Var}[\widehat{\mathbf{F}}_r(p)] = m^{-2r} \sum_{t=0}^{r-1} m^{r+t} \binom{r}{t} r^{r-t} \mathbf{F}_{r+t}(p)$.*

The estimator $\widehat{\mathbf{F}}_r(p)$ draws $\mathrm{Poi}(m)$ samples from $p$ and $m^r \cdot \widehat{\mathbf{F}}_r(p)$ equals the number of $r$-wise collisions, i.e., ordered $r$-tuples of samples that land in the same bin. We use this fact to get a few useful algorithms for approximating these moments:

**Lemma 2.2.** *There exists an algorithm that given an integer $r \geq 1$ and sample access to a distribution $p$ returns a positive real number $x$ so that:*

1. *With at least $99\%$ probability $x$ is within a constant (depending on $r$) multiple of $\|p\|_r$.*

2. *The expectation of $1/x$ is $O_r(1/\|p\|_r)$.*

3. *The expected number of samples taken by the algorithm is $O_r(1/\|p\|_r)$.*

*Proof.* The algorithm is as follows:

---
**Algorithm 1** Algorithm for Rough Moment Estimation
---
1: **procedure** ROUGH-MOMENT-ESTIMATOR$(p, r)$
**input:** Sample access to distribution $p$ on unknown discrete domain $\mathbf{\Omega}$ and an integer $r > 0$.
**output:** A value $x$ approximating $\|p\|_r$.
2:      Draw samples from $p$ until there is some $r$-wise collision among these samples.
3:      Return $1/n$, where $n$ is the number of samples taken in Step 2.

---

Firstly, we note that with large constant probability $n \gg_r 1/\|p\|_r$. This is because after taking $m$ samples, the expected number of $r$-wise collisions is at most $F_r(p)m^r = (\|p\|_r m)^r$. Thus, by Markov's inequality, if $m \ll 1/\|p\|_r$, then with large constant probability, our algorithm will not have terminated yet. To finish the proof, it suffices to show that $\mathbf{E}[n] = O_r(1/\|p\|_r)$. This implies by Markov's inequality that with large constant probability $n \ll_r 1/\|p\|_r$, and bounds the expectations of the number of samples and of $1/x$. Let $m = 1/\|p\|_r$. We note, by Fact 2.1 , if we take $\mathrm{Poi}(m)$ samples from $p$, the expected number of $r$-wise collisions is 1, and the variance is $O_r(1)$. By the Paley-Zygmund inequality, every time the algorithm takes $\mathrm{Poi}(m)$ samples, there is at least a $c_r > 0$ probability of seeing an $r$-wise collision. Therefore, if we consider our algorithm to take samples in blocks of size $\mathrm{Poi}(m)$, the probability that we have not found an $r$-wise collision after $t$ blocks is at most $(1 - c_r)^t$. Thus, the expected number of blocks until we have an $r$-wise collision is $O_r(1)$. Therefore, the expected number of samples is $O_r(m) = O_r(1/\|p\|_r)$ completing the proof. $\qquad \square$

From the above, we derive an algorithm that approximates $\|p\|_r$ to a small relative error:

**Lemma 2.3.** *There exists an algorithm that given sample access to a distribution $p$, a positive integer $r$ and a $1 > \delta > 0$, computes a value $\widehat{\gamma}_r$ so that with probability at least 19/20 we have that $|\widehat{\gamma}_r - \mathbf{F}_r(p)| \leq \delta \cdot \mathbf{F}_r(p)$. Furthermore, this algorithm uses an expected $O_r(\frac{1}{\delta^2 \|p\|_r})$ samples.*

*Proof.* The algorithm is as follows:

---
**Algorithm 2** Algorithm for Moment Estimation
---
1: **procedure** MOMENT-ESTIMATOR$(p, r, \delta)$
**input:** Sample access to arbitrary distribution $p$ on unknown discrete domain $\mathbf{\Omega}$ and an integer $r > 0$, and a $1 > \delta > 0$.
**output:** A value $\widehat{\gamma}_r$ approximating $\mathbf{F}_r(p)$.
2:      Run Rough-Moment-Estimator$(p, r)$ returning a value $x$.
3:      Let $m$ be $C_r/(\delta^2 x)$ for $C_r$ a sufficiently large constant in terms of $r$.
4:      Run the algorithm from Fact 2.1 using $\mathrm{Poi}(m)$ samples and return the result.

---

To show correctness, first note that with $99\%$ probability we have that $x = \Theta_r(\|p\|_r)$, and thus, $m$ is at least a sufficiently large multiple of $1/\delta^2 \|p\|_r$. If this holds, then the output of our algorithm will be a random variable with mean $\mathbf{F}_r(p)$. We need to bound the variance, which we do as follows:

**Claim 2.4.** *If $m\|p\|_r \gg 1$, then $\mathbf{Var}(\widehat{F_r}(p)) = O_r(\mathbf{F}_r(p)^2(\|p\|_r/m))$.*

*Proof.* The variance is $O_r\left(\sum_{t=0}^{r-1} m^{t-r}\|p\|_r^{t+r}\right) = O_r(m^{-1}\|p\|_r^{2r-1}) = O_r(\mathbf{F}_r(p)^2(\|p\|_r/m))$, which completes the proof. $\qquad \square$

If $C_r$ is large enough, this implies that $\mathbf{Var}(\widehat{\mathbf{F}}_r(p)) \le (\mathbf{F}_r(p)^2 \delta^2)/100$. Given this, our bound on $|\widehat{\gamma}_r - \mathbf{F}_r(p)|$ follows from Chebyshev's inequality. In terms of sample complexity, we note that the expected number of samples in Step 1 is $O_r(1/\|p\|_r)$, and the expected number of samples in Step 2 is $O(m) = O_r(1/(\delta^2 x))$, which in expectation is $O_r(1/(\delta^2 \|p\|_r))$. This completes the proof. $\qquad\square$

Our algorithm will begin by running Rough-Moment-Estimator to compute rough estimates of the second and third moments of $p$. Unless there is some $n$ for which $\|p\|_2 = \Theta(n^{-1/2})$ and $\|p\|_3 = \Theta(n^{-2/3})$, then we know that $p$ cannot possibly be uniform. Otherwise, we know that if $p$ is uniform, then its support must have size $\Theta(n)$. Our algorithm will thus critically depend on the following proposition:

**Proposition 2.5.** *There exists an algorithm that given sample access to a distribution $p$, and $n, \epsilon > 0$ takes an expected $O(n^{2/3}/\epsilon^{4/3} + n^{1/2}/\epsilon^2)$ samples from $p$ and distinguishes with probability at least $2/3$ between the cases: (i) $p$ is the uniform distribution on a domain of size $\Theta(n)$, and (ii) $p$ is $\epsilon$-far from any uniform distribution.*

Our algorithm will begin by verifying that $\|p\|_2 = \Theta(n^{-1/2})$ and $\|p\|_3 = \Theta(n^{-2/3})$ using Lemma 2.2. Thus, in the second case, we can assume that $\|p\|_2 = \Theta(n^{-1/2})$ and $\|p\|_3 = \Theta(n^{-2/3})$. We will further split our algorithm into cases depending on whether $\epsilon$ is bigger than $n^{-1/4}$, which in particular determines which term dominates the sample complexity.

We will need the following simple claim giving a useful condition for the soundness case:

**Claim 2.6.** *If $d_{\mathrm{TV}}(p, \mathcal{C}_U) \ge \epsilon$, then for all $x \in [0,1]$ we have that $\sum_{i \in \mathbf{\Omega}} \min\{p_i, |x - p_i|\} \ge \epsilon/2$.*

*Proof.* Let $S_h$ be the set of $i \in \mathbf{\Omega}$ on which $p_i > x/2$. Let $\delta = \sum_{i \in \mathbf{\Omega}} \min\{p_i, |x - p_i|\}$. Note that $\delta = \|p - c_{x, S_h}\|_1$, where $c_{x, S_h}$ is the pseudo-distribution that is $x$ on $S_h$ on $0$ elsewhere. If $\|c_{x, S_h}\|_1$ were $1$, $c_{x, S_h}$ would be the uniform distribution $\mathbf{u}_{S_h}$ and we would have $\delta \ge \epsilon$. However, this need not be the case. That said, it is easy to see that $\|\mathbf{u}_{S_h} - c_{x, S_h}\|_1 = |1 - \|c_{x, S_h}\|_1| \le \|p - c_{x, S_h}\|_1 = \delta$. Therefore, by the triangle inequality $2\delta \ge \|p - c_{x, S_h}\|_1 + \|\mathbf{u}_{S_h} - c_{x, S_h}\|_1 \ge \|p - \mathbf{u}_{S_h}\|_1 \ge \epsilon$. $\qquad\square$

## 2.2 Algorithm for Large $\epsilon$

**Lemma 2.7.** *There exists an algorithm that given sample access to a distribution $p$, and $n, \epsilon > 0$ with $\epsilon \ge n^{-1/4}$ takes an expected $O(n^{2/3}/\epsilon^{4/3})$ samples from $p$ and distinguishes with probability at least $9/10$ between the cases: (i) $p$ is the uniform distribution on a domain of size $\Theta(n)$. (ii) $p$ satisfies $\|p\|_2 = \Theta(n^{-1/2})$, $\|p\|_3 = \Theta(n^{-2/3})$, and $p$ is $\epsilon$-far from any uniform distribution.*

The basic idea of this algorithm is that if $p$ is uniform over any discrete domain then

$$\mathbf{F}_3(p) = \mathbf{F}_2(p)^2 . \tag{1}$$

We claim that this condition is robust. Namely for $p$ far from uniform, Equation (1) will fail by a lot. Therefore, we can distinguish between the relevant cases by finding suitably close approximations to $\mathbf{F}_2(p)$ and $\mathbf{F}_3(p)$. To start with, we need to prove the robust version of Equation (1):

**Lemma 2.8.** *We have the following: (i) If $p \in \mathcal{C}_U$, then $\mathbf{F}_3(p) = \mathbf{F}_2^2(p)$. (ii) If $d_{\mathrm{TV}}(p, \mathcal{C}_U) \ge \epsilon$, then $\mathbf{F}_3(p) - \mathbf{F}_2^2(p) > \epsilon^2 \mathbf{F}_2^2(p)/64$.*

*Proof.* The proof of (i) is straightforward. Suppose that $p = \mathbf{u}_S$ for some $\emptyset \ne S \subseteq \mathbf{\Omega}$. It then follows that $\mathbf{F}_2(p) = 1/|S|$ and $\mathbf{F}_3(p) = 1/|S|^2$, yielding part (i) of the lemma. We now proceed to prove part (ii). Suppose that $d_{\mathrm{TV}}(p, \mathcal{C}_U) \ge \epsilon$. First, it will be useful to rewrite the quantity $\mathbf{F}_3(p) - \mathbf{F}_2^2(p)$ as follows:

$$\mathbf{F}_3(p) - \mathbf{F}_2^2(p) = \sum_{i \in \mathbf{\Omega}} p_i (p_i - \mathbf{F}_2(p))^2 . \tag{2}$$

Note that (2) follows from the identity $p_i(p_i - \mathbf{F}_2(p))^2 = p_i^3 + p_i \mathbf{F}_2(p)^2 - 2p_i^2 \mathbf{F}_2(p)$ by summing over $i \in \mathbf{\Omega}$. Since $d_{\mathrm{TV}}(p, \mathcal{C}_U) \ge \epsilon$, an application of Claim 2.6 for $x = \mathbf{F}_2(p) \in [0,1]$, gives that $\sum_{i \in \mathbf{\Omega}} \min\{p_i, |\mathbf{F}_2(p) - p_i|\} \ge \epsilon/2$ . We partition $\mathbf{\Omega}$ into the sets $S_l = \{i \in \mathbf{\Omega} \mid p_i < \mathbf{F}_2(p)/2\}$ and its complement $S_h = \mathbf{\Omega} \setminus S_l$. Note that $\sum_{i \in \mathbf{\Omega}} \min\{p_i, |\mathbf{F}_2(p) - p_i|\} = \sum_{i \in S_l} p_i + \sum_{i \in S_h} |\mathbf{F}_2(p) -$

$p_i|$ . It follows that either $\sum_{i\in S_l} p_i \geq \epsilon/4$ or $\sum_{i\in S_h} |\mathbf{F}_2(p) - p_i| \geq \epsilon/4$. We analyze each case separately. First, suppose that $\sum_{i\in S_l} p_i \geq \epsilon/4$. Using (2) we can now write

$$\mathbf{F}_3(p) - \mathbf{F}_2^2(p) \geq \sum_{i\in S_l} p_i(p_i - \mathbf{F}_2(p))^2 > (\mathbf{F}_2(p)/2)^2 \cdot \sum_{i\in S_l} p_i = \epsilon\mathbf{F}_2^2(p)/16 \ .$$

Now suppose that $\sum_{i\in S_h} |\mathbf{F}_2(p) - p_i| \geq \epsilon/4$. Note that $1 \leq |S_h| \leq 2/|\mathbf{F}_2(p)|$. In this case, using (2) we obtain

$$
\begin{aligned}
\mathbf{F}_3(p) - \mathbf{F}_2^2(p) \ &\geq\ \sum_{i\in S_h} p_i(p_i - \mathbf{F}_2(p))^2 \geq (\mathbf{F}_2(p)/2) \cdot \sum_{i\in S_h} (p_i - \mathbf{F}_2(p))^2 \\
&\geq\ (\mathbf{F}_2(p)/2) \cdot \frac{(\sum_{i\in S_h} |\mathbf{F}_2(p) - p_i|)^2}{|S_h|} \geq (\mathbf{F}_2(p)/2)^2 \cdot (\epsilon/4)^2 = \epsilon^2\mathbf{F}_2^2(p)/64 \ ,
\end{aligned}
$$

where the second inequality uses the definition of $S_h$, and the third is Cauchy-Schwarz. $\qquad \square$

We are now ready to prove Lemma 2.7. At a high level, the algorithm is simple. Compute approximations to $\mathbf{F}_2(p)$ and $\mathbf{F}_3(p)$ using Fact 2.1 and apply Lemma 2.8. However, there is one technical problem with this scheme. Namely that the variance in our estimator for $\mathbf{F}_3(p)$ depends on the values of $\mathbf{F}_4(p)$ and $\mathbf{F}_5(p)$. If either of these are too large, then it will affect the accuracy of our final estimator. However, if $p$ is uniform on a domain of size $\Theta(n)$, it must be the case that $\mathbf{F}_4(p) = O(n^{-3})$ and $\mathbf{F}_5(p) = O(n^{-4})$. Se we will first perform a pre-processing step where we verify that neither $\mathbf{F}_4(p)$ nor $\mathbf{F}_5(p)$ are too large, before estimating $\mathbf{F}_2(p)$ and $\mathbf{F}_3(p)$.

*Proof of Lemma 2.7.* The pseudocode is described in Algorithm 3.

---

**Algorithm 3** Algorithm for Large $\epsilon$

1: **procedure** LARGE-EPS-TESTER$(p, n, \epsilon)$
**input:** Sample access to arbitrary distribution $p$ on unknown discrete domain $\mathbf{\Omega}$ and $n, \epsilon > 0$ and $\epsilon \geq n^{-1/4}$.
**output:** "YES" with probability $9/10$ if $p$ is uniform on a set of size $\Theta(n)$, "NO" with probability $9/10$ if $\|p\|_2 = \Theta(n^{-1/2})$ and $\|p\|_3 = \Theta(n^{-2/3})$ and $p$ is $\epsilon$-far from any uniform distribution.
2:      Let $C, C'$ be a sufficiently large constants, with $C$ large enough relative to $C'$. Let $m = Cn^{2/3}/\epsilon^{4/3}$.
3:      Draw Poi$(O(m))$ samples from $p$ and let $\widehat{\gamma}_4$ denote the value of $\widehat{\mathbf{F}}_4(p)$ on this sample.
4:      **if** $\widehat{\gamma}_4 > C'n^{-3}$ **then return** "NO".
5:      Draw Poi$(O(m))$ samples from $p$ and let $\widehat{\gamma}_5$ denote the value of $\widehat{\mathbf{F}}_5(p)$ on this sample.
6:      **if** $\widehat{\gamma}_5 > C'n^{-4}$ **then return** "NO".
7:      Compute the estimates $\widehat{\mathbf{F}}_2(p)$, $\widehat{\mathbf{F}}_3(p)$ on two separate sets of Poi$(m)$ samples.
8:      **if** $\left( \widehat{\mathbf{F}}_3(p) - \widehat{\mathbf{F}}_2(p)^2 \leq \epsilon^2/(300n^2) \right)$ **then return** "YES".
9:      **else return** "NO".

---

Note that the expected number of samples taken by this algorithm is $O(m) = O(n^{2/3}/\epsilon^{4/3})$. We next prove correctness. We start by considering Steps 3 through 6. Firstly, in the completeness case, we note that $\mathbf{F}_r(p) = \Theta(n^{1-r})$, and therefore, by the Markov bound, $\widehat{\gamma}_r \leq C'n^{1-r}$ with at least 99% probability. In the completeness case, we claim that these steps will reject with at least 99% probability unless $\mathbf{F}_r(p) = O(C'(n^{1-r} + m^{-r}))$. In particular, if $\mathbf{F}_r(p) \geq KC'(n^{1-r} + m^{-r})$, then $m\|p\|_r \geq 1$, and therefore, by Claim 2.4 we have that $\mathbf{E}[\widehat{\gamma}_r] = \mathbf{F}_r(p)$ and $\mathbf{Var}(\widehat{\gamma}_r) = O(\mathbf{F}_r(p)^2/K^2)$. So, if $K$ is sufficiently large, by Chebyshev's inequality, with 99% probability we have that $(\gamma)_r > \mathbf{F}_r(p)/2 \geq C'n^{1-r}$. Thus, in the remainder, we can assume that $\mathbf{F}_4(p) = O(C'(n^{-3} + m^{-4}))$ and $\mathbf{F}_5(p) = O(C'(n^{-4} + m^{-5}))$. To analyze Step 7, we note that $\mathbf{Var}(\widehat{F_2}(p)) = O(m^{-2}\mathbf{F}_2(p) + m^{-1}\mathbf{F}_3(p)) = O(m^{-2}n^{-1} + m^{-1}n^{-2}) = O(\epsilon^4/n^2)/C$ , where we use that $\epsilon \geq n^{-1/4}$ and $m = Cn^{2/3}/\epsilon^{4/3}$. Similarly, we have

$$
\begin{aligned}
\mathbf{Var}(\widehat{F_3}(p)) &= O(m^{-3}\mathbf{F}_3(p) + m^{-2}\mathbf{F}_4(p) + m^{-1}\mathbf{F}_5(p)) \\
&= O(m^{-3}n^{-2} + C'm^{-2}n^{-3} + C'm^{-6} + C'm^{-1}n^{-4}) = O(\epsilon^4/n^4)(C'/C).
\end{aligned}
$$

Therefore, by Chebyshev's inequality, with 99% probability we have that $|\widehat{F_2}(p) - \mathbf{F}_2(p)| = O(\epsilon^2/n)/\sqrt{C}$, and $|\widehat{F_3}(p) - \mathbf{F}_3(p)| = O(\epsilon^2/n^2)\sqrt{C'/C}$. Assuming these hold, we have that

$$\left| \left(\mathbf{F}_3(p) - \mathbf{F}_2(p)^2\right) - \left(\widehat{\mathbf{F}_3}(p) - \widehat{\mathbf{F}_2}(p)^2\right) \right| = O(\epsilon^2/n^2)\sqrt{C'/C}.$$

Thus, if $C/C'$ is sufficiently large, if $p$ is uniform, we accept, and if $p$ is $\epsilon$-far from uniform, then by Lemma 2.8, we reject. This completes the proof. $\qquad\square$

## 2.3 Algorithm for Small $\epsilon$

In this section, we give a tester that works for $\epsilon \leq n^{-1/4}$.

**Lemma 2.9.** *There exists an algorithm that given sample access to a distribution $p$, and $n, \epsilon > 0$ with $\epsilon \leq n^{-1/4}$ takes an expected $O(n^{1/2}/\epsilon^2)$ samples from $p$ and distinguishes with probability at least $9/10$ between the cases: (i) $p$ is the uniform distribution on a domain of size $\Theta(n)$, and (ii) $p$ is $\epsilon$-far from any uniform distribution.*

*Proof.* The basic idea is that we will take $\Theta(n)$ samples from $p$ and let $S$ be the set of distinct elements seen. We then test uniformity of $(p|S)$ using the standard uniformity tester.

---

**Algorithm 4** Algorithm for Small Epsilon

1: **procedure** SMALL-EPS-TESTER$(p, n, \epsilon)$
**input:** Sample access to arbitrary distribution $p$ on unknown discrete domain $\mathbf{\Omega}$ and $n, \epsilon > 0$ and $n^{-1/4} \geq \epsilon$.
**output:** "YES" with probability $9/10$ if $p$ is uniform on a set of size $\Theta(n)$, "NO" with probability $9/10$ $p$ is $\epsilon$-far from any uniform distribution.
2:      Let $C, C'$ be a sufficiently large constants with $C$ large even relative to $C'$. Let $m = Cn$.
3:      Draw Poi$(m)$ samples from $p$. Let $S$ be the subset of $\mathbf{\Omega}$ that appears in the sample.
4:      Verify the following conditions: (i) Each $i \in S$ appears $O(C \log n)$ times; (ii) $|S| = \Theta(n)$.
5:      **if** (either of conditions (i) or (ii)) is violated **then return** "NO".
6:      Draw $m' = C\sqrt{n}/\epsilon^2$ samples from $p$.
7:      **if** fewer than half of these samples were in $S$ **then return** "NO".
8:      Use the first $m'/2$ of these samples that landed in $S$ to run the standard uniformity tester for $(p|S)$ with distance $\epsilon/C'$ and 1% probability of error.
9:      **return** the answer of the tester in Step 8.

---

We note that the expected number of samples is $O(m + m') = O(n^{2/3}/\epsilon^{4/3})$. It remains to prove correctness. We begin with the completeness case. If $p$ is uniform over a set of size $\Theta(n)$, with high probability no bin will see more than $O(C \log(n))$ samples, thus (i) is satisfied. Furthermore, we note that with high probability that Poi$(Cn)$ samples from $p$ will cover more than two thirds of the bins with high probability and thus (ii) will be satisfied. Additionally, this means that $p(S) \geq 2/3$, so again with high probability, at least half of our $m'$ samples will lie in $S$. These first $m'/2$ samples from $S$ will be independent samples from $(p|S)$, which is uniform, and therefore with 99% probability will pass the uniformity tester. Therefore, in this case, our algorithm will return "YES" with probability at least $9/10$.

For the soundness case, we note that if any bin has probability more than a sufficiently large multiple of $\log(n)/n$, we will fail to satisfy (i) with high probability and reject. We would like to claim next that $(p|S)$ is likely to be far from uniform, and thus that we will fail the final test. Of course, this may depend on the randomness over our first set of samples, but we claim it with high probability. In particular, we show (see supplementary material for the proof):

**Lemma 2.10.** *If $d_{\mathrm{TV}}(p, \mathcal{C}_U) \geq \epsilon$ and $p$ assigns no more than $O(\log(n)/n)$ mass to any single bin, then with high probability over the $\mathrm{Poi}(m)$ samples, we have at least one of the following: (i) $|S|$ is not $\Theta(n)$, (ii) $p(S) \leq 1/3$, (iii) $d_{\mathrm{TV}}((p|S), \mathcal{C}_U) \geq \epsilon/C'$.*

---
**Algorithm 5** The Full Tester
---
1: **procedure** GENERALIZED-UNIFORMITY-TESTER$(p, \epsilon)$
**input:** Sample access to arbitrary distribution $p$ on unknown discrete domain $\mathbf{\Omega}$ and $n, \epsilon > 0$.
**output:** "YES" with probability $2/3$ if $p$ is uniform on its support, "NO" with probability $2/3$ $p$ is $\epsilon$-far from any uniform distribution.
2:     Let $\widehat{\gamma_2} = $ Rough-Moment-Estimator$(p, 2)$.
3:     Let $\widehat{\gamma_3} = $ Rough-Moment-Estimator$(p, 3)$.
4:     **if** $\widehat{\gamma_3}$ is not $\Theta(\widehat{\gamma_2}^{4/3})$ **then return** "NO".
5:     Let $n = \widehat{\gamma_3}^{-3/2}$.
6:     **if** $\epsilon \geq n^{-1/4}$ **then return** Large-Eps-Tester$(p, n, \epsilon)$
7:     **if** $n^{-1/4} \geq \epsilon$ **then return** Small-Eps-Tester$(p, n, \epsilon)$
---

## 2.4 Full Tester

First, we verify correctness. With appropriately high probability, $\widehat{\gamma_2}$ and $\widehat{\gamma_3}$ approximate $\|p\|_2$ and $\|p\|_3$ respectively to within constant factors. In this case, $p$ cannot be uniform unless $\widehat{\gamma_3} = \Theta(\widehat{\gamma_2}^{4/3})$. Assuming this holds, $\mathbf{F}_2(p) = \Theta(n^{-1/2})$ and $\mathbf{F}_3(p) = \Theta(n^{-2/3})$, so the assumptions necessary for our Small/Large-$\epsilon$ testers are satisfied, and they will work with appropriate probability.

For sample complexity, we note that the first two lines take $O(1/\|p\|_3)$ samples in expectation. The remaining lines use an expected $O(n^{2/3}/\epsilon^{4/3} + n^{1/2}/\epsilon^2)$ samples. This is $O(1/(\epsilon^{4/3}\widehat{\gamma_3}) + 1/(\epsilon^2\widehat{\gamma_2}))$. Our final expected sample bound follows from noting by Lemma 2.2 that the expected values of $1/\widehat{\gamma_3}$ and $1/\widehat{\gamma_2}$ are $O(1/\|p\|_3)$ and $O(1/\|p\|_2)$, respectively. This completes our proof.

## 3 Sample Complexity Lower Bound

In this section, we sketch a sample size lower bound matching our algorithm in Proposition 2.5. One part of the lower bound is fairly easy. In particular, it is known [Pan08] that $\Omega(\sqrt{n}/\epsilon^2)$ samples are required to test uniformity of a distribution with a known support of size $n$. It is easy to see that the hard cases for this lower bound still work when $\|p\|_2 = \Theta(n^{-1/2})$ and $\|p\|_3 = \Theta(n^{-2/3})$.

The other half of the lower bound is somewhat more difficult and we rely on the lower bound techniques of [DK16]. In particular, for $n > 0$, and $1/10 > \epsilon > n^{-1/4}$ and for $N$ sufficiently large, we produce a pair of distributions $\mathcal{D}$ and $\mathcal{D}'$ over positive measures on $[N]$, so that: 1. A random sample from $\mathcal{D}$ or $\mathcal{D}'$ has total mass $\Theta(1)$ with high probability. 2. A random sample from $\mathcal{D}$ or $\mathcal{D}'$ has $\|p\|_2 = \Theta(n^{-1/2})$ and $\|p\|_3 = \Theta(n^{-2/3})$ with high probability. 3. A sample from $\mu \in \mathcal{D}$ has $\mu/\|\mu\|_1$ the uniform distribution over some subset of $[N]$ with probability 1. 4. A sample from $\mu \in \mathcal{D}'$ has $\mu/\|\mu\|_1$ at least $\Omega(\epsilon)$-far from any uniform distribution with high probability. 5. Given a measure $\mu$ taking randomly from either $\mathcal{D}$ or $\mathcal{D}'$, no algorithm given the output of a Poisson process with intensity $k\mu$ for $k = o(\min(n^{2/3}/\epsilon^{4/3}, n))$ can reliably distinguish between a $\mu$ taken from $\mathcal{D}$ and $\mu$ taken from $\mathcal{D}'$.

Before we exhibit these families, we first discuss why the above is sufficient. This Poissonization technique has been used previously in various settings [VV14, DK16, WY16, DGPP17], so we only provide a sketch here. In particular, suppose that we have such families $\mathcal{D}$ and $\mathcal{D}'$, but that there is also an algorithm $A$ that distinguishes between a distribution $p$ being uniform and being $\epsilon$-far from uniform when $\|p\|_2 = \Theta(n^{-1/2})$ and $\|p\|_3 = \Theta(n^{-2/3})$ in $m = o(n^{2/3}/\epsilon^{4/3})$ samples. We show that we can use algorithm $A$ to violate property 5 above. In particular, letting $p = \mu/\|\mu\|_1$ for $\mu$ a random measure taken from either $\mathcal{D}$ or $\mathcal{D}'$, we note that with high probability $\|p\|_2 = \Theta(n^{-1/2})$ and $\|p\|_3 = \Theta(n^{-2/3})$. Therefore, $m' = o(n^{2/3}/\epsilon^{4/3})$ samples are sufficient to distinguish between $p$ being uniform and being $\Omega(\epsilon)$ far from uniform. However, by properties 3 and 4, this is equivalent to distinguish between $\mu$ being taken from $\mathcal{D}$ and being taken from $\mathcal{D}'$. On the other hand, given the output of a Poisson process with intensity $Cm'\mu$, for $C$ a sufficiently large constant, a random $m'$ of these samples (note that there are at least $m'$ total samples with high probability) are distributed identically to $m'$ samples from $p$. Thus, applying $A$ to these samples distinguishes between $\mu$ taken from $\mathcal{D}$ and $\mu$ taken from $\mathcal{D}'$, thus contradicting property 5. Due to space constraints, the technical details are deferred to the supplementary material.

## 4 Conclusions

In this paper, we gave tight upper and lower bounds on the sample complexity of generalized uniformity testing – a natural non-trivial generalization of uniformity testing, recently introduced in [BC17]. The obvious research question is to understand the sample complexity of testing more general symmetric properties (e.g., closeness, independence, etc.) for the regime where the domain of the underlying distributions is discrete but unknown (of unknown size). Is it possible to obtain sub-learning sample complexities for these problems? And what is the optimal sample complexity for each of these tasks?

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
