[Supplementary Material · generalized-uniformity-neurips-11-14.pdf]

# Appendix

## A  Proof of Lemma 2.10

Suppose that $d_{\mathrm{TV}}(p, \mathcal{C}_U) \geq \epsilon$. We want to show that with high probability over the samples it holds $\sum_{i \in S} |p_i - p(S)/|S|| = \Omega(\epsilon)$. The main difficulty is that the value of $p(S)$ is unknown, hence we need a somewhat indirect argument. By Claim 2.6, for all $x \in [0, 1]$ we have that

$$\sum_{i \in \boldsymbol{\Omega}} \min\{p_i, |p_i - x|\} \geq \epsilon/2 . \tag{3}$$

To show that $Z(x) \stackrel{\text{def}}{=} \sum_{i \in S} |p_i - x| = \Omega(\epsilon)$, when $x = p(S)/|S|$. To do this, we note that for any $S$, $Z(x)$ always attains a minimum at $p_i$ for some $i$. Furthermore, if $|S| = \Theta(n)$ and $p(S) \geq 1/3$, then $Z(x)$ is automatically large unless $x = \Theta(1/n)$. Thus, it suffices to show that:

**Claim A.1.** *With probability at least $19/20$, for all $x = p_i = \Theta(1/n)$, we have that $Z(x) = \Omega(\epsilon)$.*

*Proof.* We note that there are only $O(n)$ allowable values of $x$, and so we will prove that for any given $x = \Theta(1/n)$ that the statement holds with high probability.

Let $Z_i$, $i \in \boldsymbol{\Omega}$, be the indicator of the event $i \in S$. Then, $Z(x) = \sum_{i \in \boldsymbol{\Omega}} |p_i - x| Z_i$. Note that $Z_i$ is a Bernoulli random variable with $\mathbf{E}[Z_i] = 1 - e^{-p_i m}$ and that the $Z_i$'s are mutually independent. Note that $\mathbf{E}[Z(x)] = \sum_{i \in \boldsymbol{\Omega}} (1 - e^{-p_i m}) |p_i - x|$. We recall the following concentration inequality for sums of non-negative random variables (see, e.g., Exercise 2.9 in [BLM13]):

**Fact A.2.** *Let $X_1, \ldots, X_k$ be independent non-negative random variables, and $X = \sum_{j=1}^{k} X_j$. Then, for any $t > 0$, it holds that $\Pr[X \leq \mathbf{E}[X] - t] \leq \exp\left(-t^2/(2\sum_{i=1}^{k} \mathbf{E}[X_i^2])\right)$.*

Since $Z(x) = \sum_{i \in \boldsymbol{\Omega}} |p_i - x| Z_i$ where the $Z_i$'s are independent Bernoulli random variables with $\mathbf{E}[Z_i^2] = 1 - e^{-p_i m}$, an application of Fact A.2 yields that

$$\Pr\left[Z(x) \leq \mathbf{E}[Z(x)] - t\right] \leq \exp\left(\frac{-t^2}{2\sum_{i \in \boldsymbol{\Omega}}(1 - e^{-p_i m})(p_i - x)^2}\right) . \tag{4}$$

Let $S_l = \{i \in \boldsymbol{\Omega} : p_i \leq x/2\}$ and $S_h = \boldsymbol{\Omega} \setminus S_l$. By (3), we get that $\sum_{i \in S_l} p_i + \sum_{i \in S_h} |x - p_i| \geq \epsilon/2$ . For $i \in S_l$, we have that $(1 - e^{-p_i n})|p_i - x| \geq m \cdot p_i \cdot |x/2| = \Omega(p_i)$. For $i \in S_h$, we have that $(1 - e^{-p_i n}) = \Omega(1)$ and therefore $(1 - e^{-p_i m})|p_i - x| = \Omega(1)|p_i - x|$. We therefore get that $\mathbf{E}[Z(x)] = \Omega(\epsilon)$. We now bound $\sum_{i \in \boldsymbol{\Omega}}(1 - e^{-p_i m})(p_i - x)^2$ from above using the fact that $p_i = O(\log n/n)$, for all $i \in \boldsymbol{\Omega}$. This assumption and the range of $x$ imply that

$$\sum_{i \in \boldsymbol{\Omega}} (1 - e^{-p_i m})(p_i - x)^2 \leq O(\log n/n) \cdot \mathbf{E}[Z] .$$

So, by setting $t = \mathbf{E}[Z]/2$ in (4), we get that

$$\Pr[Z(x) \leq \mathbf{E}[Z(x)]/2] \leq \exp\left(-\Omega(\epsilon n/\log n)\right) = \exp\left(-n^{\Omega(1)}\right) ,$$

where the last inequality follows from the range of $\epsilon$. Recalling that there are only $O(1/n)$ many allowable values of $x$, Claim A.1 follows by a union bound. $\square$

Lemma 2.10 follows from noting that it suffices to show that $Z(p(S)/|S|) = \Omega(\epsilon)$ when $|S| = \Theta(n)$ and $p(S) \geq 1/3$. In such a case, $Z(x)$ takes a minimum when $x = p_i$ for some $i$. If $x = \Theta(1/n)$, the result follows from our claim. Otherwise, it is easy to see that $Z(x) = \Omega(1)$ for all $x$ not $\Theta(1/n)$. This completes the proof of our lemma. $\square$

To complete our analysis of the soundness case, we have that unless $p$ assigns some bin probability $\Omega(\log(n)/n)$, that with high probability over samples, either we are rejected by (ii), have $p(S) < 1/3$ or $(p|S)$ is $\Omega(\epsilon)$-far from uniform. If $p(S) \leq 1/3$, most of our $m'$ samples lie outside of $S$ with high probability. If $(p|S)$ is far from uniform, our $m'$ samples from Step 6 either mostly lie outside of $S$ (in which case we reject), or the first $m'/2$ of them are independent random samples from $(p|S)$. Since $(p|S)$ is $\epsilon/C'$-far from uniform, our uniformity tester will reject with $99\%$ probability. This completes our proof.

# B   Omitted Proofs from Section 3

We exhibit the relevant families $\mathcal{D}$ and $\mathcal{D}'$. In both cases, we want to arrange $\mu_i := \mu(\{i\})$ to be i.i.d. for different $i$. We also want it to be the case that the first and second moments of $\mu_i$ are the same for $\mathcal{D}$ and $\mathcal{D}'$. Combining this with requirements on closeness to uniform, we are led to the following definitions:

For $\mu$ taken from $\mathcal{D}'$, we let

$$\mu_i = \begin{cases} \frac{1+\epsilon}{n} & \text{, with probability } \frac{n}{2N} \\ \frac{1-\epsilon}{n} & \text{, with probability } \frac{n}{2N} \\ 0 & \text{, otherwise .} \end{cases}$$

For $\mu$ taken from $\mathcal{D}$, we let

$$\mu_i = \begin{cases} \frac{1+\epsilon^2}{n} & \text{, with probability } \frac{n}{N(1+\epsilon^2)} \\ 0 & \text{, otherwise .} \end{cases}$$

Note that in both cases, the average total mass is 1, and it is easy to see by Chernoff bounds that the actual mass of $\mu$ is $\Theta(1)$ with high probability. Additionally, in both cases the expected sizes of $\|p\|_2^2$ and $\|p\|_3^3$ are $\Theta(n^{-1})$ and $\Theta(n^{-2})$, respectively. Again, it is not hard to show by a Chernoff bound that with high probability the actual second and third moments of $p$ are within constant factors of this. For $\mu$ taken from $\mathcal{D}$, all of the $\mu_i$ are either 0 or $\frac{1+\epsilon^2}{n}$, and thus $\mu/\|\mu\|_1$ is uniform over its support. For $\mu$ taken from $\mathcal{D}'$, with high probability at least a third of the bins in its support have $\mu_i = \frac{1+\epsilon}{n}$, and at least a third have $\mu_i = \frac{1-\epsilon}{n}$. If this is the case, then at least a constant fraction of the mass of $\mu/\|\mu\|_1$ comes from bins with mass off from the average mass by at least a $(1 \pm \epsilon)$ factor, and this implies that $\mu/\|\mu\|_1$ is at least $\Omega(\epsilon)$-far from uniform.

We have thus verified 1-4. Property 5 will be somewhat more difficult to prove. For this, let $X$ be a random $\{0, 1\}$ random variable with equal probabilities. Let $\mu$ be chosen randomly from $\mathcal{D}$ if $X = 0$, and randomly from $\mathcal{D}'$ if $X = 1$. Let our Poisson process with intensity $k\mu$ return $A_i$ samples from bin $i$. We note that, by the same arguments as in [DK16], it suffices to show that the shared information $I(X; A_1, \ldots, A_N) = o(1)$. In order to prove this, we note that the $A_i$ are conditionally independent on $X$, and thus we have that $I(X; A_1, \ldots, A_N) \leq \sum_{i=1}^N I(X; A_i) = NI(X; A_1)$. Thus, we need to show that $I(X; A_1) = o(1/N)$. For notational simplicity, we drop the subscript in $A_1$.

This boils down to an elementary but tedious calculation. We begin by noting that we can bound

$$I(X; A) = \sum_{t=0}^{\infty} O\left( \frac{(\Pr(A = t | X = 0) - \Pr(A = t | X = 1))^2}{\Pr(A = t)} \right) .$$

(This calculation is standard. See Fact 81 in [CDKS17] for a proof.) We seek to bound each of these terms. The distribution of $A$ conditioned on $\mu_1$ is Poisson with parameter $k\mu_1$. Thus, the distribution of $A$ conditioned on $X$ is a mixture of two or three Poisson distributions, one of which is the trivial constant 0. We start by giving explicit expressions for these probabilities.

Firstly, for the $t = 0$ term, note that

$$\Pr(A = t | X = 1) = 1 - \frac{n}{N}\left( 1 - \frac{e^{-k(1+\epsilon)/n} + e^{-k(1-\epsilon)/n}}{2} \right) ,$$

$$\Pr(A = t | X = 0) = 1 - \frac{n}{N(1+\epsilon^2)}\left( 1 - e^{-k(1+\epsilon^2)/n} \right) .$$

Note that $\Pr(A = 0)$ is at least $1 - n/N \geq 1/2$ and $\Pr(A = t | X = 1) - \Pr(A = t | X = 0) \leq n/N$. Thus, the contribution from this term, $\frac{(\Pr(A=0|X=0) - \Pr(A=0|X=1))^2}{\Pr(A=0)}$, is $O(n/N)^2 = o(1/N)$.

For $t \geq 1$, there is no contribution from $\mu_1 = 0$. We can compute the probabilities involved exactly as

$$\Pr(A = t | X = 1) = \frac{n}{N} \frac{(k(1+\epsilon)/n)^t e^{-k(1+\epsilon)/n} + (k(1-\epsilon)/n)^t e^{-k(1-\epsilon)/n}}{2t!} ,$$

$$\Pr(A = t | X = 0) = \frac{n}{N(1 + \epsilon^2)} \frac{(k(1 + \epsilon^2)/n)^t e^{-k(1+\epsilon^2)/n}}{t!} ,$$

and obtain that $\frac{(\Pr(A=t|X=0)-\Pr(A=t|X=1))^2}{\Pr(A=t)}$ is

$$O\left(\left(\frac{n^{1-t}k^t}{2Nt!}\right) \frac{\left((1+\epsilon)^t e^{-k(1+\epsilon)/n} + (1-\epsilon)^t e^{-k(1-\epsilon)/n} - 2(1+\epsilon^2)^{t-1} e^{-k(1+\epsilon^2)/n}\right)^2}{(1+\epsilon)^t e^{-k(1+\epsilon)/n} + (1-\epsilon)^t e^{-k(1-\epsilon)/n} + 2(1+\epsilon^2)^{t-1} e^{-k(1+\epsilon^2)/n}}\right) .$$

Factoring out the $e^{-k/n}$ terms and noting that, since $k\epsilon/n = o(1)$, the denominator is $\Omega(e^{-k/n})$ yields that

$$O\left(\left(\frac{n^{1-t}k^t e^{-k/n}}{2Nt!}\right)\left((1+\epsilon)^t e^{-k(1+\epsilon)/n} + (1-\epsilon)^t e^{-k(1-\epsilon)/n} - 2(1+\epsilon^2)^{t-1} e^{-k(1+\epsilon^2)/n}\right)^2\right) .$$

Noting that $k/n = o(1)$, we can ignore this $e^{-kn}$ term and Taylor expanding the exponentials, we have that

$$\frac{(\Pr(A = t | X = 0) - \Pr(A = t | X = 1))^2}{\Pr(A = t)} =$$

$$O\left(\left(\frac{n^{1-t}k^t}{2Nt!}\right)\left((1+\epsilon)^t(1 - k(1+\epsilon)/n) + (1-\epsilon)^t(1 + k(1-\epsilon)/n)\right.\right.$$

$$\left.\left. - 2(1+\epsilon^2)^{t-1}(1 - k(1+\epsilon^2)/n) + O((k\epsilon/n)^2(1+\epsilon)^t))\right)^2\right) .$$

We deal separately with the cases $t = 1, t = 2$ and $t > 2$. For the $t = 1$ term, we have

$$O\left(\left(\frac{k}{N}\right)\left((1+\epsilon)(1 - k\epsilon/n) + (1-\epsilon)(1 + k\epsilon/n) - 2(1 - k\epsilon^2/n) + O((k\epsilon/n)^2)\right)^2\right)$$

$$=O\left(\left(\frac{k}{N}\right)O((k\epsilon/n)^2)^2\right) .$$

Since $k = o(n^{2/3}/\epsilon^{4/3})$ and $\epsilon > n^{-1/4}$, $\epsilon k/n = o(n^{-1/3}/\epsilon^{1/3}) = o(n^{-1/4})$, and we find that this is

$$O\left(\left(\frac{k}{N}\right)o(1/n)\right) = o(1/N) .$$

This appropriately bounds the contribution from this term.

When $t = 2$, we have

$$O\left(\left(\frac{k^2}{nN}\right)\left((1+\epsilon)^2(1 - k(1+\epsilon)/n) + (1-\epsilon)^2(1 - k(1-\epsilon)/n)\right.\right.$$

$$\left.\left. -2(1+\epsilon^2)(1 - k(1+\epsilon^2)/n) + O((k\epsilon/n)^2)\right)^2\right) .$$

Note that the terms without $k/n$ factors cancel out, $(1+\epsilon)^2 + (1-\epsilon)^2 - 2(1+\epsilon^2) = 0$, yielding

$$O(k^2/nN)(k\epsilon^2/n + o(n^{-1/2}))^2 = O(k^4\epsilon^4/n^3 N) + o(k^2/n^2 N) = o(k^3\epsilon^4/n^2 N) + o(1/N) = o(1/N) ,$$

using both $k = o(n^{2/3}/\epsilon^{4/3})$ and $k = o(n)$.

For $t > 2$, we let $f_t(x) = (1 + x)^t(1 - kx/n)$. In terms of $f_t$, we have that $\frac{(\Pr(A=t|X=0)-\Pr(A=t|X=1))^2}{\Pr(A=t)}$ is:

$$O\left(\left(\frac{n^{1-t}k^t}{2Nt!}\right)(f_t(\epsilon) + f_t(-\epsilon))/2 - f_t(0) - (f_{t-1}(\epsilon^2) - f_{t-1}(0)) + o(n^{-1/2})^2\right) .$$

Using the Taylor expansion of $f_t$ in terms of its first two derivatives and $f_{t-1}$ in terms of its first, we see that

$$(f_t(\epsilon) + f_t(-\epsilon))/2 - f_t(0) = \epsilon^2 f_t''(\xi)$$

and
$$f_{t-1}(\epsilon^2) - f_{t-1}(0) = \epsilon^2 f'_{t-1}(\xi') \ ,$$

for some $\xi \in [-\epsilon, \epsilon]$ and $\xi' \in [0, \epsilon^2]$. However, the derivatives are
$$f'_t(x) = (1+x)^{t-1}(t - (1+x+tx)k/n)$$

and
$$f''_t(x) = (1+x)^{t-2}(t(t-1) - t(t+1)xk/n) \ ,$$

and so $|f''_t(\xi)| \le O(t^2(1+\epsilon)^{t-1})$ and $f'_{t-1}(\xi') \le O(t(1+\epsilon^2)^{t-2})$. Hence, the term
$$\frac{(\Pr(A = t|X = 0) - \Pr(A = t|X = 1))^2}{\Pr(A = t)}$$

is at most
$$\begin{aligned} O(n^{1-t}k^t/Nt!)&(\epsilon^4 t^4(1+\epsilon)^{2t-2}) + o(1/n)) \\ &= O\left((k^3\epsilon^4/n^2)(t^4(1+\epsilon)^2/N)(k(1+\epsilon)^2/n)^{t-3}/t!\right) + o\left((k/n)^t/(Nt!)\right) \\ &= o(1/N)t^4/t! \ , \end{aligned}$$

using both $k = o(n^{2/3}/\epsilon^{4/3})$ and $k = o(n)$. Since $(t+1)^4/(t+1)! \le t^4/2t!$ for all $t \ge 4$, even summing the above over all $t \ge 3$ still leaves $o(1/N)$.

Thus, we have that $I(X; A) = o(1/N)$, and therefore that $I(X : A_1, \ldots, A_N) = o(1)$. This proves that $X = 0$ and $X = 1$ cannot be reliably distinguished given $A_1, \ldots, A_N$, and thus proves property 5, completing the proof of our lower bound.