[Reviews · NeurIPS 2018]

Reviewer 1



This paper considers the recently introduced problem of generalized uniformity testing where the goal is to test if a distribution is uniform on a support of unknown size or is it \epsilon-far in total variation distance from all uniform distributions. The main result of the paper is a complete characterization of sample complexity of this problem. It is shown that this sample complexity is \Theta( 1/(||p||_3 \epsilon^{4/3}) +1/(||p||_2\epsilon^{2}) ), when the unknown distribution is p. Note that when p is uniform on n, this sample complexity becomes \Theta( n^{2/3}/\epsilon^{4/3}) +n^{1/2}/\epsilon^{2})), which exceeds \sqrt{n}/\epsilon^{2} (the sample complexity of the problem when the support of the distribution is known to be n) if n^{1/4}>1/\epsilon. The algorithm achieving this bound, first uses estimates for ||p||_2 and ||p||_3 to ensure that they satisfy ||p||_2^4 = ||p||_3^3 (which must be the case if the distribution p is uniform) and define n= 1/||p||_2^2. Then, it proceeds to accomplish uniformity testing assuming that the support is n. When p is indeed uniform, it seems that any standard uniformity test for support-size n can be used and its analysis will go through. However, the technical difficulty arises from the alternative where we might be off in our estimate of support. The main contribution of this paper is to identify appropriate statistic that is robust to this uncertainty. Specifically, the paper presents a different algorithm for the small \epsilon (n^{1/4}<1/\epsilon) and large \epsilon (n^{1/4}>1/\epsilon) regimes. For large epsilon, it evaluates ||p||_3^3-||p||_2^4 and compares it with \epsilon^2/n^2. An interesting component of analysis of this case shows how to control the variance, which depends on ||p||_4, ||p||_5. Note that it is not possible to estimate these quantities with the available sample budget. Yet, the proof shows that we can ensure the desired bound on the quantities within our sample budget; an extra check is included in the algorithm to ensure from estimates of ||p||_4, ||p||_5 that the variance is small as desired. For small epsilon, we have the usual sample budget of O(n^{1/2}/\epsilon^2). But it is not clear if the usual collision based test can be used here since we don’t have a handle on its variance when p is not uniform. Instead, the authors propose a different test which first takes a sample of length n and ensure that all symbols appear at least log n times. Note that in this regime we can draw n samples since n

Reviewer 2



The authors consider the problem of generalized uniformity testing. This problem -- which is a natural extension to the standard uniformity testing problem -- was recently been proposed and studied by Batu and Canonne (2017). In standard uniformity testing problem, the task is to distinguish the case where the samples are coming from (the given) uniform density, or are being generated from a probability measure that is epsilon-far from the mentioned uniform density (in terms of total variation distance). In generalized uniformity testing, however, the tester should still accept if the distribution is uniform over a "subset" of the domain. As observed and proved by Batu and Canonne (2017), the sample-complexity bounds for generalized uniformity testing exhibit a somewhat surprising divergence from those of uniformity testing. In this submission, the authors extend the results of Batu and Canonne (2017) by improving both the lower bound and the upper bound (for the sample complexity), essentially characterizing the sample complexity of generalized uniformity testing. Furthermore, the provided algorithm is computationally efficient. The improvement in the lower bound is achieved using the standard lower bound for uniformity testing, as well as the use of the poissonization technique which is known for proving lower bounds (still, using the poissonization technique requires a non-trivial construction). The improvement in the upper bound is achieved using multiple delicate improvements over the work of Batu and Canonne (2017), which is on a high level based on comparing the second and the third moments. Overall, I think this a solid theoretical result about an interesting and important problem. The ideas are also well described. I think for this conference, it would make sense to add some more motivations about why generalized uniformity testing is important (e.g., use cases)

Reviewer 3



This works studies the problem of generalized uniformity testing that was recently introduced by Batu and Canonne (FOCS'17). In this problem, one is given sample access to a distribution over a discrete domain and is asked to determine if the distribution is uniform over some subset of the domain or far in total variation distance from any such distribution. Batu and Canonne showed that O(1)/(|p|_3 eps^6) samples suffice to test up to total variation distance eps, where |p|_3 is the L3 norm of the underlying distribution which is n^(2/3) for a uniform distribution on n elements. The main result of this paper is pinning down the exact sample complexity of the problem. In particular, it is shown that 1/(|p|_3 eps^(4/3)) + 1/(|p|_2 eps^2) samples are both necessary and sufficient for this problem. The upper bound works by examining two cases based on the magnitude of eps. It tests which of the terms 1/(|p|_3 eps^(4/3)) and 1/(|p|_2 eps^2) is larger and uses different strategies for each of the cases. To distinguish between them, it uses a method to (roughly) approximate within constant factors the L2 and L3 norms of the distributions by counting collisions. In the first case where eps is large, the algorithm works by checking whether |p|_3^3 - |p|_2^4 is 0 or not by computing more fine-grained estimations of the norms. A key lemma that enables this algorithm is that the difference is 0 for a uniform distribution and sufficiently far from 0 when the distribution is far from uniform. The second case where eps is small is easier, as one can afford to recover the support of the distribution. In that case, a standard uniformity tester is performed on the recovered support. In addition to the upper bounds, the paper shows a lower bound of 1/(|p|_3 eps^(4/3)) + 1/(|p|_2 eps^2). For the term 1/(|p|_2 eps^2), the standard uniformity testing lower-bound by Paninski applies here. In that lower bound, the distribution is either uniform or (1+eps)/n over a random half of the support and (1-eps)/n on the other half. To get the lower bound for the term 1/(|p|_3 eps^(4/3)), this work uses a similar construction over a random subset of the whole domain. To establish the lower-bound, a Poissonization argument is used together with the method of shared information of Diakonikolas and Kane (FOCS'16). Overall, I think that this is a very interesting work settling the sample complexity of a very well motivated and natural problem. It improves both upper and lower bounds significantly over the previous work, giving very simple and natural algorithms for testing. The paper is well written and self contained with very intuitive explanations and presents many techniques in simple arguments behind the proofs. I enjoyed reading the paper and support acceptance.